

# Estimating the influence of transport to aerosol size distributions during new particle formation events

Runlong Cai[1], Indra Chandra[2,3], Dongsen Yang[4], Lei Yao[5], Yueyun Fu[1], Xiaoxiao Li[1], Yiqun Lu[5], Lun Luo[6], Jiming Hao[1], Yan Ma[4], Lin Wang[5], Jun Zheng[4], Takafumi Seto[2], Jingkun Jiang[1,*]

[1]State Key Joint Laboratory of Environment Simulation and Pollution Control, School of Environment, Tsinghua University, Beijing 100084, China
[2]Department of Chemical and Material Engineering, Kanazawa University, Kanazawa 920-1192, Japan
[3]Engineering Physics, School of Electrical Engineering, Telkom University 40261, Indonesia
[4]Collaborative Innovation Center of Atmospheric Environment and Equipment Technology, Nanjing University of Information Science & Technology, Nanjing 210044, China
[5]Shanghai Key Laboratory of Atmospheric Particle Pollution and Prevention (LAP3), Department of Environmental Science and Engineering, Fudan University, Shanghai 200433, China
[6]South-East Tibetan plateau Station for integrated observation and research of alpine environment (SETS), Institute of Tibetan Plateau Research, Chinese Academy of Sciences, Beijing 100101, China

*Correspondence to*: Jingkun Jiang (jiangjk@tsinghua.edu.cn)

**Abstract.** New particle formation (NPF) and the subsequent particle growth occur frequently in various atmospheric environments. Significant influence of transport on aerosol size distributions is commonly observed, especially for non-regional NPF events. With certain assumptions and approximations, a population balance method is proposed to examine the influence of transport on the temporal evolution of aerosol size distributions during NPF events. The method is derived from the aerosol general dynamic equation in the continuous form. Meteorological information (e.g., wind speed, wind direction, water vapor mixing ratio) was used to complement the analysis. The NPF events observed in South-East Tibet, Fukue Island, and urban Beijing were analyzed using the proposed method. Significant contribution of transport to the observed aerosol size distributions is found during the NPF events in both South-East Tibet and Fukue Island. The changes in the contribution of transport is in good correlation with the changes in wind speed and direction. This correlation indicates that local mountain and valley breezes govern the observed new particles at the South-East Tibet site. Most NPF events observed at Fukue Island are closely related to the long-range transport of aerosols and gaseous precursors due to the movement of air masses. Regional NPF events are typically observed in urban Beijing and the contribution of transport to the observed aerosol size distributions is negligible compared to condensational growth and coagulation scavenging. In relatively clean atmospheric environment, the proposed method can be used to characterize the contribution of transport to particles in the size range from ~10 nm to ~50 nm. However, during intense NPF events in relatively polluted atmosphere, the estimated contribution of transport is sensitive to the uncertainties in condensational growth and coagulation scavenging due to the dominance of their corresponding terms in the population balance equation.



# 1 Introduction

Understanding the formation mechanism of secondary aerosols and their interactions with clouds and radiation will contribute to a better prediction of global climate (IPCC, 2013). Secondary new particle formation (NPF) is a common phenomenon observed in various atmospheric environments (Kulmala et al., 2004). During an NPF event, gaseous precursors form molecular clusters spontaneously by random collisions, which can grow up rapidly into larger particles due to condensation if they survive from evaporation and coagulation scavenging. Both field measurements and theoretical modelling indicate that NPF is an important source of cloud condensation nuclei (CCN, Kuang et al., 2009; Kerminen et al., 2012). During a typical regional NPF event, new particles can be formed in a spatial scale extending over several hundred kilometers (Hussein et al., 2009), thus nucleation and the subsequent growth of particles are usually regarded as regional phenomena and the influence of transport is often assumed to be negligible (e.g., Kulmala et al., 2001; Cai and Jiang, 2017). However, during a non-regional NPF event, the influence of transport can be significant.

The shape of temporal evolution of aerosol size distributions was used to identify whether the observed NPF event is a regional one or not. A banana-shaped event was usually regarded as a regional event (Kulmala et al., 2012), whereas other shapes (e.g., "apple" and "hump" shapes) may indicate non-regional events (Vana et al., 2008). For instance, significant increases in sub-50 nm particle concentrations were observed at ground level during precipitation in the central Amazon basin due to the vertical transport from the lower free troposphere to the boundary layer (Wang et al., 2016). At Jungfraujoch in Switzerland, the increase in the number concentration of 10-50 nm particles was observed on 55% of days without any direct evidence of local nucleation or condensational growth. The observed new particles were perhaps formed off-site (Tröstl et al., 2016). At Mace Head on the west coast of Ireland, gaseous precursors are formed away from the observation site and thus "apple" and "hump" events were sometimes observed (Vana et al., 2008). In fact, reported influence of transport on particle formation can be dated back to John Aitken's studies (e.g., Aitken, 1912). Usually, it is easy to identify that the observed aerosol size distributions are non-negligibly affected by transport during these "non-banana" events.

In addition, a typical "banana" event does not evidently indicate that new particles are formed without the influence of transport. During some "banana" NPF events, dramatic increase in the number concentration of sub-5 nm particles was not observed, indicating that the observed new particles were not formed locally. For instance, only particle growth was observed without high concentrations of sub-5 nm particles during some NPF events at Fukue Island. A possible reason is that nucleation occurred upstream of the observation site and the newly formed particles had grown into larger diameters during the long-range-transport of the air masses (Chandra et al., 2016). The contribution of transport to the observed aerosol size distributions cannot be simply assumed negligible in such events. To the best of our knowledge, however, methods to quantitatively analyze a NPF event for estimating the contribution of transport have not been reported previously.

The population balance method based on the aerosol general dynamic equation may help to estimate the influence of transport. In previous studies, the population balance method was used to estimate new particle formation rates (Kulmala et al., 2001; Kulmala et al., 2012; Cai and Jiang, 2017) and size-resolved particle growth rate (Kuang et al., 2012). Neglecting the influence



of transport is one of the fundamental assumptions of these applications. In contrast, if condensational growth and particle loss due to coagulation scavenging can be properly quantified, the contribution of transport to the observed aerosol size distributions can perhaps be estimated.

In this study, we propose a method for examining the contribution of transport on the temporal evolution of the observed aerosol size distributions during atmospheric NPF events. It is derived from the aerosol general dynamic equation in the continuous form and the population balance assumption with certain assumptions and approximations. The proposed method is applied to analyze the NPF events observed in South-East Tibet, Fukue Island, and urban Beijing. Meteorological data at these sites are also analyzed. The feasibility and uncertainties of the proposed method are discussed.

## 2 Theory

The proposed method for estimating the contribution of transport to the temporal evolution of aerosol size distributions is based on the aerosol general dynamic equation. For a particle larger than ~10 nm that consists of sufficient molecules, the particle diameter can be approximated to be a continuous (rather than discrete) function of the molecular number. Accordingly, we use the general dynamic equation in the continuous form to characterize the temporal evolution of aerosol size distributions in this study. The general dynamic equation accounting for the influence of transport is shown in Eq. 1:

$$\frac{\mathrm{d}N_{[i,j]}}{\mathrm{d}t} = GR_i n_i - GR_j n_j + CoagSrc_{[i,j]} - CoagSnk_{[i,j]} + TR_{[i,j]}, \tag{1}$$

where the subscripts i and j correspond to the specific particle diameters ($d_p$, in m), $d_i$ and $d_j$, respectively; $N_{[i,j]}$ (in $\#\cdot m^{-3}$) is the number concentration of particles ranging from $d_i$ to $d_j$; $t$ is time (in s); $\mathrm{d}N/\mathrm{d}t$ characterizes the change in the observed particle number concentration (in $\#\cdot m^{-3}\cdot s^{-1}$); $GR$ is the condensational growth rate (in $m\cdot s^{-1}$) defined as $\mathrm{d}d_p/\mathrm{d}t$; $n$ is the aerosol size distribution function (in $\#\cdot m^{-4}$), $\mathrm{d}N/\mathrm{d}d_p$; $CoagSrc_{[i,j]}$ and $CoagSnk_{[i,j]}$ are the formation and loss rates due to coagulation for particles in the size range from $d_i$ to $d_j$ (in $\#\cdot m^{-3}\cdot s^{-1}$), respectively; and $TR_{[i,j]}$ is the newly introduced transport term (in $\#\cdot m^{-3}\cdot s^{-1}$, explained below). The terms on the right-hand side of Eq. 1 correspond to the five processes leading to the change in observed particle number concentration: particle condensational growth into and out of the size range, formation and scavenging due to coagulation, and the contribution of transport. The theoretical expressions for $CoagSrc_{[i,j]}$ and $CoagSnk_{[i,j]}$ in the integral form are shown in Eq. 2 and Eq. 3, respectively (Kuang et al., 2012).

$$CoagSrc_{[i,j]} = \frac{1}{2}\int_{d_i}^{d_j}\int_{0}^{d_y} \beta_{(x,\bar{x})} n_x n_{\bar{x}} \frac{d_y^2}{d_{\bar{x}}^2} \mathrm{d}d_x \mathrm{d}d_y \tag{2}$$

$$CoagSnk_{[i,j]} = \int_{d_i}^{d_j}\int_{0}^{+\infty} \beta_{(x,y)} n_x n_y \mathrm{d}d_x \mathrm{d}d_y \tag{3}$$

$\beta_{(x,y)}$ is the coagulation coefficient ($m^3\cdot s^{-1}$) for particles with the diameter $d_x$ and those with the diameter $d_y$. The value of $\beta_{(x,y)}$ can be estimated using the Fuchs' formula (Eq. 13.56, Seinfeld and Pandis, 2006). $d_x$ and $d_y$ are variables representing particle





diameters determined by the limits of integration. $\bar{x}$ is the subscript of $d_{\bar{x}}$ and $d_{\bar{x}}$ is defined by $d_x^3 + d_{\bar{x}}^3 = d_y^3$. $n_x$, $n_y$, and $n_{\bar{x}}$ are the aerosol size distribution functions ($dN/dd_p$) at $d_x$, $d_y$, and $d_{\bar{x}}$, respectively. The term $d_y^2/d_{\bar{x}}^2$ comes from $d_{\bar{x}}^2 \cdot dd_{\bar{x}} = d_y^2 \cdot dd_y$.

The transport term in Eq. 1, $TR_{[i,j]}$, characterizes the contribution of transport to the observed $dN_{[i,j]}/dt$. However, the physical

meanings of $TR_{[i,j]}$ and $dN_{[i,j]}/dt$ should be specially clarified. When aerosol is well mixed, i.e., there is no difference among the aerosol size distributions at different spatial positions, condensational growth and coagulation are the only causes of the observed $dN_{[i,j]}/dt$. However, there is no absolute homogeneity of aerosols in the real ambient air. Since the air is constantly flowing whereas the sampling inlet for the instrument is usually fixed at a specific position, the observed aerosol size distributions at different time actually correspond to different aerosol populations. As shown in Fig. 1, the observed $N_{[i,j]}$ at

time $t$ is $N_{[i,j]}(t, z_1)$ and $z_1$ denotes the detected aerosol population at $t$. For the sake of illustration, we may assume that the aerosols at different spatial positions do not mix with each other and hence $z_1$ can be related to the Lagrangian position of the observed aerosol population. After a short time interval, $dt$, we assume that the observed particle number concentration is $N_{[i,j]}(t+dt, z_2)$ where $z_2$ indicates that the detected aerosol population has changed due to transport. The measured time rate of change of particle number concentration, $dN_{[i,j]}/dt$ is the mutual result of changes in both time and fluid position, which can

be mathematically described as:

$$\frac{dN_{[i,j]}}{dt} = \lim_{dt \to 0} \frac{N_{[i,j]}(t+dt, z_2) - N_{[i,j]}(t, z_1)}{dt}. \tag{4}$$

At time $t+dt$, the number concentration of aerosols at position $z_1$ should become $N_{[i,j]}(t+dt, z_1)$. The value of $N_{[i,j]}(t+dt, z_1)$ can be theoretically estimated using the general dynamic equation:

$$\lim_{dt \to 0} \frac{N_{[i,j]}(t+dt, z_1) - N_{[i,j]}(t, z_1)}{dt} = GR_i n_i - GR_j n_j + CoagSrc_{[i,j]} - CoagSnk_{[i,j]}, \tag{5}$$

Equations 1, 4, and 5 indicate that $TR_{[i,j]}$ essentially characterizes the difference between the measured and the theoretically predicted aerosol concentrations, i.e., the difference between the two aerosol populations, $z_1$ and $z_2$, as shown in Eq. 6. Note that the illustrations above maintains generality without the assumption of aerosol mixing except that there is no simple relationship between $z$ and the Lagrangian position of aerosols when considering mixing.

$$TR_{[i,j]} = \lim_{dt \to 0} \frac{N_{[i,j]}(t+dt, z_2) - N_{[i,j]}(t+dt, z_1)}{dt} \tag{6}$$

$TR_{[i,j]}$ can be estimated using Eq. 1 if the other terms can be properly quantified with certain assumptions. The formulas to estimate $CoagSrc_{[i,j]}$ and $CoagSnk_{[i,j]}$ can be simplified if the concerned size range, $[d_i, d_j]$ is relatively narrow. $CoagSnk_{[i,j]}$ can be approximated using the following equation,

$$CoagSnk_{[i,j]} = N_{[i,j]} \cdot \sum_{k=1}^{max} \beta_{(m,k)} N_k \tag{7}$$





The subscript m corresponds to the representative diameter, $d_m$ of the size range $[d_i, d_j]$. $d_m$ is usually the arithmetic or geometric mean diameter of $d_i$ and $d_j$. $N_k$ is the particle number concentration in a size bin and k is the corresponding bin number. $\beta_{(m, k)}$ is the coagulation coefficient between the two particles with diameters $d_m$ and $d_k$. The summation term $\sum_{k=1}^{max} \beta_{(m,k)} N_k$ is the coagulation sink (*CoagS*, Kulmala et al., 2001), however, *CoagS* is not referred to below to avoid

confusion with *CoagSnk*. Note that when using Eq. 7 to estimate the *CoagSnk*[i,j], the concerned size range, $[d_i, d_j]$ should be narrow so that $d_m$ can serve as a representative diameter when estimating the coagulation coefficients.

During NPF events in relatively clean atmospheric environments, *CoagSrc*[i,j] is usually much smaller than *CoagSnk*[i,j] for nucleation mode particles (< 50 nm). As shown in Eqs. 2 and 3, one of the integral for *CoagSnk*[i,j] covers all the particle diameters whereas *CoagSrc*[i,j] is only determined by particles in a limited size range. In addition, the coagulation coefficient,

$\beta_{(x, y)}$ tends to be higher if the difference between $d_x$ and $d_y$ are larger. Thus, it is reasonable to simply neglect *CoagSrc*[i,j] during NPF events in relatively clean atmospheric environments (where the number concentration of new particles is relatively low) like in some previous balance formulas to estimate new particle formation rates (Kulmala et al., 2001; Kulmala et al., 2012).

In the perspective of the kinetic theory of gases, the condensational increment in the particle diameter is caused by random collisions of various gaseous precursors (such as sulfuric acid and organics) onto pre-existing particle surfaces. Without loss

of generality, we may use a single representative gaseous precursor, e.g., sulfuric acid, to illustrate the size-dependence of the growth rate. The growth rate contributed by the condensation of sulfuric acid monomer can be estimated as follows (Weber et al., 1997; Kuang et al., 2010),

$$GR_{SA} = \frac{V_{SA} N_{SA} \bar{v} \beta_m}{2} \times \frac{\left(d_p + d_{SA}\right)^2}{d_p^2}, \tag{8}$$

where $V_{SA}$ is the volume of a sulfuric acid monomer (in $m^3$); $N_{SA}$ is the number concentration of sulfuric acid monomers (in

$\# \cdot m^{-3}$); $\bar{v}$ is the mean relative thermal velocity between a sulfuric acid monomer and a particle (in m/s); $\beta_m$ is a fitted dimensionless correction coefficient; and $d_{SA}$ is the diameter of a sulfuric acid monomer (in m). An expression for $\beta_m$ is shown in Eq. 8 (Fuchs and Sutugin, 1971), where $\alpha$ is the mass accommodation coefficient and $Kn$ is the Knudsen number.

$$\beta_m = \frac{\alpha \times \left(Kn + Kn^2\right)}{\alpha + \left(1.33 + 0.38\alpha\right) Kn + 1.33 Kn^2} \tag{9}$$

The condensational growth rates for particles in a certain size range (e.g., 10-50 nm) can be approximated as a constant value.

The size-dependence of particle grow rate is relatively weak in the size range from 10-50 nm. When assuming an $\alpha$ of 1.0, the estimated growth rate for 50 nm particles using Eq. 8 is ~82% of the grow rate for 10 nm particles (293 K, 101 kPa). Note that the value of $GR_{SA}$ is insensitive to $d_{SA}$ in the size range from 10-50 nm, i.e., the diameters of different gaseous precursors do not significantly affect the size-dependency of the condensational growth rate. The evaporation of gaseous precursors due to the Kelvin effect is significant for small particles, especially in the sub-3 nm size range. However, it can be neglected for





relatively large particles. Size-dependent growth rates were estimated during atmospheric NPF events (Kuang et al., 2012) and in chamber studies (Pichelstorfer et al., 2018). The estimated growth rates were relatively constant within acceptable uncertainties when the particles were larger than certain diameters, e.g., ~3 nm in Kuang et al. (2012) and ~10 nm in Pichelstorfer et al. (2018). Thus, $GR_i$ and $GR_j$ in Eq. 1 can be approximated using a size-independent $GR$ inferred from the

measured aerosol size distributions (e.g., Kulmala et al., 2012). Note that the approximation of the size-independence is only valid for a limited size range, e.g., 10-50 nm, yet such a size range is usually sufficiently wide for analysis on the influence of transport during typical NPF events.

Based on Eq. 1 and the discussions above, the formula to estimate the influence of particle transport in practical applications can be simplified as:

$$TR_{[i,j]} = \frac{\mathrm{d}N_{[i,j]}}{\mathrm{d}t} + CoagSnk_{[i,j]} + GR(n_j - n_i) \ , \qquad (10)$$

where the corresponding diameters of i and j ($d_i$ and $d_j$) are usually the lower and upper size limits of each particle size bin, respectively. The last term in Eq. 10 is referred as the growth term in the following discussions. $\mathrm{d}N_{[i,j]}/\mathrm{d}t$ and $CoagSnk_{[i,j]}$ can be estimated using the particle size distributions and $GR$ is inferred from the time rate of change of the fitted peak diameter (see Fig. 3a). The contribution of transport on the apparent increase in particle diameter is neglected in this study. A negative

growth term indicates that more particles grow into than out of the concerned size bin, $[d_i, d_j]$. Although $CoagSrc_{[i,j]}$ is neglected in Eq. 10, the contribution of the coagulation source may be partially accounted for because the observed particle growth is the mutual result of condensation and coagulation (Stolzenburg et al., 2005).

For the particles in the concerned size range, $[d_i, d_j]$, the assumptions (indicated by •) and approximations (indicated by ○) made in Eq. 10 are listed as follows:

- the influence of primary emission, particle evaporation, and other losses except for coagulation loss are negligible;
- condensation of gaseous vapors is the reason of the observed particle growth;
- the condensational growth rates for different particle sizes in the concerned size range are the same;
- the mass accommodation coefficient is assumed to be known (1.0 in this study);
○ particle formation due to coagulation ($CoagSrc_{[i,j]}$) is negligible compared to particle loss due to coagulation

scavenging ($CoagSnk_{[i,j]}$) and condensational growth (characterized using the growth term);
○ a representative particle diameter can be used when estimating the coagulation coefficients (see Eq. 7).

One should check the reliability of these assumptions before using Eq. 10 in case of the influence of other potential particle sources or sinks. It should be clarified that the transport term, $TR_{[i,j]}$ is simply a mathematically defined parameter using the balance method rather than directly derived from first principles. Equation 10 does not guarantee that the estimated transport

term is the result of transport. In fact, the transport term indicates all the neglected physical processes and the errors of other estimated terms in Eq. 1. Thus, additional information such as meteorological data should be used to infer the source of the transport term. For instance, when the changing height of the boundary layer is the major reason for the change in particle number concentration (although the timescale of this change is much larger than typical timescale of the variations in aerosol



size distributions during NPF events), the estimated transport term mainly characterizes the effect of dilution rather than convectional transport. Equation 10 is usually not valid for analyzing sub-5 nm particles (where 5 nm is only an approximate value rather than an critical threshold) because the condensational growth rate may be strongly size-dependent in this range (Kuang et al., 2012; Kulmala et al., 2013; Pichelstorfer et al., 2018). For particles larger than ~200 nm, liquid phase reactions

are possibly an important mechanism for particle growth (McMurry and Wilson, 1983), indicating significant errors in assuming a size-independent growth-rate in this size range. In addition, if there is a significant error in the method to estimate the growth rate (Li and McMurry, 2018), the error will also propagate in the estimated transport term.

The approximations above (indicated by ○) can also be violated in some circumstances. In this case, Eq. 11 can be readily updated using Eqs. 2 and 3. However, the discussion in section 4.3 will reveal that the estimated transport term may be sensitive

to the uncertainties if the approximations are violated due to the uncertainties in the estimated $CoagSnk_{[i,j]}$.

## 3 Experiment

The proposed method was used to analyze aerosol size distributions observed at three sites: the South-East Tibet site, the Fukue Island site, and the urban Beijing site. The observation in South-East Tibet was conducted at the South-East Tibetan plateau Station for integrated observation and research of alpine environment (94°44' E, 29°46' N, elevation: 3326 m, Guo et al., 2016).

The station is located at Lulang river valley, Linzhi, China. The Lulang river valley is the intersection of the Lulang river and the Niyang river and is surrounded by mountains such as Gyala Peri (~23 km away, elevation: 7,294 m) and Namcha Barwa (~35 km away, elevation: 7782 m). There are ~300 people living in the nearest village located ~1.5 km away. The whole Lulang town holds ~1100 people and the town center is situated ~7 km away. In front of the station is the China National Highway 318 located ~50 m away from the sampling inlet, which was a potential anthropogenic pollution source. However,

no significant influence of traffic emission was found from the measured CO concentration, the chemical composition of $PM_1$, and the aerosol size distributions.

The field campaign was conducted from Jul. 15[th] to Sep. 3[rd], 2016, which was during the monsoon season in the Lulang river valley. The precipitations were frequent in July and gradually descended since August. The influence of pollutant long-range transport is the lowest during this season among the whole year (Zhao et al., 2013). The aerosol size distributions on total 38

days were recorded during the campaign while the rest data was missing because of the instability of electricity supply or the malfunction of the local hydropower station. A particle size distribution (PSD) system including two scanning mobility particle spectrometers (SMPSs) covering different size range and one aerodynamic particle sizer (APS, TSI 3321) was used to measure particles ranging from 3 nm to 10 μm (Liu et al., 2016). The sub-3 nm aerosol size distributions were measured using a prototype diethylene glycol SMPS (DEG-SMPS, Cai et al., 2017a) and a commercialized particle size magnifier (PSM,

Airmodus A11). The laminar condensation particle counters (CPCs) serving as the detectors of the SMPSs probably underestimated the particle concentrations around the 50% cut-off size due to the low ambient pressure of 69 kPa (Takegawa et al., 2016). Thus, the data for 2-6 nm particles of the incorporated aerosol size distributions were from the DEG-SMPS and



the sub-2 nm data measured by the DEG-SMPS did not partcipate in any calculation. The PSM might also be affected by the inlet pressure problem (Kangasluoma et al., 2016). The meteorological data were obtained from the local weather station (Guo et al., 2016), including wind speed, wind direction, temperature, relative humidity, and the atmospheric pressure.

The Fukue Island Supersite (32°48' N, 128°42' E) is located on the Fukue Island, Nagasaki, Japan. The site is usually on the

pathway of the long-range-transport of particles and gaseous precursors in East Asian region during winter and spring. The aerosol size distributions in the range from 3 nm to 710 nm were monitored using two SMPSs. A DEG-SMPS (TSI 3938E77) was deployed to measure sub-3 nm particles since 2016. Local wind speed and direction were measured. The backward trajectories were obtained from online HYSPLIT server of national oceanic and atmospheric administration (NOAA). Details on the Fukue Island site were reported in previous studies (Takami et al., 2005; Seto et al., 2013; Chandra et al., 2016).

The observation site in urban Beijing is located on the campus of Tsinghua University (40°00' N, 116°19' E, He et al., 2001; Cai and Jiang, 2017). The field campaign to study NPF were conducted in March and April 2016. The aerosol size distributions ranged from ~1 nm to 10 μm were measured using the PSD system and the prototype DEG-SMPS. The meteorological data was obtained from a weather station (Davis 6250) on the site. More details on the campaign can be found in Cai and Jiang (2017) and Cai et al. (2017b).

## 4 Results and discussion

### 4.1 Lulang river valley

Frequent NPF events were observed in the Lulang river valley; however, most of the new particles were not formed locally at the observation site. As the newly formed particles grow into larger sizes, the particle number concentration decreases due to the coagulation loss. Thus, the maximum number concentration of smaller new particles during an NPF event are theoretically

larger than the concentration of larger new particles. However, the daily maximum number concentration of 5-10 nm particles was always lower than the concentration of 10-50 nm particles and their ratio was mostly smaller than 40% on the observed NPF days in the Lulang river valley (Fig. 2). This low concentration of 5-10 nm particles was not sufficient to explain the origin of the observed 10-50 nm particles if assuming a regional NPF event with negligible influence of transport. For instance, on Aug. 12th, a typical NPF day, the number concentration of sub-50 nm particles increased sharply around 10:30 (Fig. 3).

These new particles grew at a relatively constant growth rate of 2.6 nm/h for ~6 hours while their number concentration decreased steadily with fluctuations during the same period (after the sharp increase to the peak value). However, the increase in particle number concentration at the beginning of the observed NPF event was mainly contributed by 10-50 nm particles, whereas a correspondingly high number concentration of 3-10 nm particles or sub-3 nm particles was not observed. To confirm that the observed low concentration of sub-3 nm particles was not caused by the instrumental detection efficiency, the DEG-

SMPS was used to classify and detect particles with a fixed diameter of 2 nm for the whole NPF day on Aug. 19th. No particles were detected at all over the whole day expect for totally 11 counts possibly due to background noise of the instrument. Although there were uncertainties in the detection efficiency of the DEG-SMPS due to the low ambient pressure (69 kPa), the





classified particle diameter was accurate because the pressure effect was accounted for. The low sub-10 nm particle concentration indicates that the majority of the observed particles were formed away from the observation site and the observed aerosol size distributions were non-negligibly affected by transport.

The mountain and valley breezes dominate the local air transport in the Lulang river valley. The valley ground is heated after sunrise and the warm air climb up along the slopes, leading to a gradual decline in the ambient pressure on the ground level (as shown in Fig. 4). The air at higher altitude transports downward vertically during daytime due to the pressure difference. The mountain breeze blows into the valley during nighttime. The circulation of the mountain and valley breezes is a regular pattern in the Lulang river valley throughout a year (Wang et al., 2010). Note that the local transport due to the mountain and valley breezes does not necessarily accompanies with the movement of air mass in a large spatial scale. As shown in Fig. 4, the water vapor mixing ratio, i.e., the amount of water vapor in per unit dry air (in g/kg), was relatively constant on Aug. 12th, indicating the air circulation occurred inside the same air mass.

The contribution of transport on the observed aerosol size distributions at different diameters on Aug. 12th was analyzed using the proposed method. The temporal evolution of particle number concentration ($dN_{[i,j]}/dt$) and the other terms in Eq. 10 were converted into the temporal evolution of aerosol size distribution function and subsequently integrated since midnight. For instance, the transformations applied to the $dN_{[i,j]}/dt$ term are shown in Eq. 11:

$$\frac{dN_{[i,j]}}{dt} \xrightarrow[\div(d_j-d_i)]{\text{divided by bin length}} \frac{dN_{[i,j]}}{dt \cdot (d_j - d_i)} \approx \frac{d\,dN/dd_m}{dt}$$

$$\frac{d\,dN/dd_m}{dt} \xrightarrow{\text{integrate since midnight}} \int_0^t \frac{d\,dN/dd_m}{dt} \cdot dt = \left.\frac{dN}{dd_m}\right|_0^t \tag{11}$$

$d_m$ is the mean diameter of $d_i$ and $d_j$. In this study, $d_i$ and $d_j$ are the limits of particle diameter for the same size bin. The $\left.dN/dd_m\right|_0^t$ term is the difference between the size distribution function ($dN/dd_p$ at $d_m$) at time $t$ and at midnight. The transport term, *CoagSnk*, and the growth term in Eq. 10 are named as cumulative terms correspondingly (e.g., cumulative transport term) after the transformations in Eq. 11, characterizing the cumulative contributions to the observed aerosol size distributions since midnight. These transformations facilitate the comparison among different size bins and help to reduce the impact of the fluctuations in the observed aerosol size distributions during field measurements. Note that a positive cumulative growth term indicates net particle growth out of the concerned size bin, $[d_i, d_j]$, while the cumulative *CoagSnk* is always non-negative.

The cumulative transport terms at different particle diameters is correlated with the shift between the mountain and valley breezes. The valley breeze started to blow around 10:30 on Aug. 12th, indicated by the changes in wind speed and direction (Fig. 4). As shown in Fig. 5, the size distribution functions ($dN/dd_p$) for 10 nm and 20 nm particles increased sharply at almost the same time and the cumulative transport term was the predominant cause. The contribution of transport to the size distribution function for 10 nm particles remained at a relatively constant level while the size distribution function decreased gradually because of coagulation scavenging and particle growth into larger sizes. The relatively large fluctuations in the observed the size distribution function and the cumulative transport term were possibly due to atmospheric turbulence. The



size distribution function for 20 nm particles showed a similar trend except for a negative contribution of transport between 12:00 and 14:00. For 30 nm and 40 nm particles, transport did not lead to a sharp increase in their size distribution functions when the local wind shifted from mountain breeze to valley breeze (around 10:30). The increases in the size distribution functions for 30 nm and 40 nm particles were mainly due to condensational growth indicated by the comparatively large cumulative growth terms. At around 17:00, the wind direction changed back from north to west, indicating the mountain breeze started to take over the control. The sudden decreases in the size distribution function and the cumulative transport term were thus observed at all the four analyzed diameters. After 19:00, the mountain breeze dominated the wind field in the valley and air stability increased (as indicated by the standard deviation of the measured wind direction, wind direction, and water vapor mixing ratio), thus the cumulative transport terms kept at relatively stable levels. There were no significant differences among the wind speed and direction measured at four different heights from 1.5 m and 18 m, indicating that turbulent mixing in the vertical direction was not a major reason for the cumulative transport term. In addition, no direct correlation was observed during the NPF events between the aerosol size distribution function and the radiation represented by the photolysis frequencies of ozone, $J(O^1D)$, indicating that radiation was not the direct cause of the rapid changes in the observed aerosol concentrations. On all the NPF days observed in the Lulang valley during the campaign, the aforementioned characteristics were observed, i.e.,: 1) the number concentration of nanoparticles increased sharply around noon, however, most of these new particles were larger than 10 nm; 2) the continuous banana-shaped particle growth was observed; 3) the number concentration of grown-up new particles decreased suddenly in the late afternoon (around 17:00). During most NPF events, the changes in the aerosol size distribution function and the cumulative transport term were correlated with the changes of the local wind field. According to the analysis using the proposed population balance method, the particle transport governed by the mountain and valley breezes contributed significantly to the observed NPF events in the Lulang Valley. We hypothesize that the new particles were formed elsewhere and brought to the site by the valley breeze. The measured sub-5 nm particle concentration was relatively low because the new particles had grown up when they arrived at the observation site. The observed new particles were probably formed upstream of the valley breeze, either in the upper air or in the valley. More studies are needed to further evaluate this hypothesis.

**4.2 Fukue Island**

The contribution of transport to the observed aerosol size distributions at Fukue Island was correlated with the changes of the air mass origin during typical NPF events. The observed NPF events at Fukue Island were classified into two types (Chandra et al., 2016) according to whether sub-10 nm particles were observed (type-A) or not (type-B). On Mar. 16th, 2015, the local wind speed and direction were relatively stable before 22:00. However, a typical type-B NPF event was observed, indicating the new particles was formed elsewhere and subsequently transported to the observation site (Fig. 6). The cumulative transport term was estimated using Eqs. 10 and 11. It should be clarified that particle loss due to coagulation was underestimated because the size distributions of particles larger than 710 nm were not measured. However, it is reasonable to assume that the underestimation was negligible because the *CoagSnk* for each size bin was mainly contributed by particles smaller than 500




nm (as shown in Fig. S1). The cumulative transport terms for 15 nm and 20 nm particles started to increase around 14:00 and the cumulative transport terms for 30 nm and 40 nm particles started to increase at around 16:00. All the estimated cumulative transport terms were relatively stable after 19:00 (Fig. 7). No significant change was observed in the wind speed or wind direction during the NPF period whereas the water vapor mixing ratio increased steadily before the NPF event, indicating that

the transport occurred in a relatively large spatial scale during the type-B NPF event. According to the backward trajectories, the origins of the air masses observed at the Fukue Island site changed gradually from the Korean Peninsula to the Yellow sea (Fig. S2) from 13:00 to 18:00. Thus, we infer that new particles was formed upstream of the air mass trajectories on the Yellow sea but probably not in the Korean Peninsula for the analyzed type-B NPF event.

The change in the wind direction is another potential cause of the local particle transport at the Fukue Island site. During the

type-A NPF event on Mar. 2nd, 2016, the estimated transport terms of 10 nm, 15 nm and 20 nm particles fluctuated between 11:00 and 15:00 (Fig. 8). No significant change was observed in the backward trajectory of the air mass during the event (Fig. S3). As shown in Fig. 8, there was a sharp valley in the transport term at approximately 12:00. At the same time, the wind direction started to change from east to the west. The wind direction at the observation site changed from north to southwest from 11:00 to 16:00, which is correlated with the cumulative transport terms. Accordingly, the estimated cumulative transport

terms for the type-A NPF event mainly reflect the inhomogeneity of the air masses at the observation site rather than their movement.

### 4.3 Urban Beijing

The proposed method was used to analyze an intensive NPF event observed in urban Beijing on Mar. 23rd, 2016. The minimum detected particle electrical mobility diameter was 1.53 nm during the event (Cai et al., 2017b). The wind direction at the

observation site was north or northwest between 8:00 and 16:00 and the wind speed was relatively stable. There was no direct evidence to prove this NPF event was a regional event, i.e., by comparing the particle size distributions observed at two different sites. However, the characteristics of the observed NPF event agrees with the criterions of the regional NPF events (Kulmala et al., 2013) and the NPF events observed in urban Beijing were usually observed simultaneously at a background station 120 km away (Wang et al., 2013). As shown in Fig. 9, the estimated cumulative transport terms for 20 nm and 30 nm

particles during the whole NPF day were negligible compared to the estimated cumulative *CoagSnk* and the cumulative growth terms. Different from the NPF events observed in clean atmospheres, the concentrations of the newly formed particles and background aerosols are comparatively high during typical NPF events in Beijing (Cai and Jiang, 2017). The *CoagSnk* and the growth term in the population balance formulas (e.g., Eqs. 1 and 10) are usually much larger than $dN/dt$. The negligible contribution of transport to the observed aerosol size distributions on Mar. 23rd agreed well with the characteristics of an

regional event (as shown in the contour plot of aerosol size distributions, Fig. 9a).

When analyzing the intense NPF events in relatively polluted atmospheric environment such as urban Beijing, however, the estimated contribution of transport is sensitive to the uncertainties in *CoagSnk* and the growth term. The transport term is obtained based on the population balance method thus the uncertainties in the estimated *CoagSnk* and growth term will




propagate in the estimated transport term. During the analyzed NPF event on Mar. 23rd, the transport term was approximately 10% that of the *CoagSnk* term and the growth term. There is a probability that the estimated transport term is mainly due to the uncertainties in estimating *CoagSnk* and the growth term rather than particle transport. For instance, a 30% overestimation of the particle growth rate will cause an approximately 300% overestimation of the transport term. The systematic errors in

the estimated transport term can be reduced using Eq. 2 and Eq. 3 instead of Eq. 7, yet the errors are still sensitive to the uncertainties in the observed aerosol size distributions. Thus, the reliability of the estimated contribution of transport is challenged when the approximations of the *CoagSnk* and the growth term in Eq. 10 (indicated by ○) are violated. Nevertheless, the analysis above still indicates a negligible contribution of transport to the observed NPF event in urban Beijing.

## 4.4 Remarks on the feasibility of the balance method

During the NPF events analyzed above, correlations were found between the estimated transport terms and the changes in the local wind field or the movement of air masses. These correlations indicate that the proposed balance method is feasible to characterize the contribution of transport to the observed aerosol size distributions. In addition to NPF events, the proposed balance method can theoretically be applied to analyze non-NPF events. However, the particle growth rate is usually difficult to determine during non-NPF periods.

One should always keep in mind that the transport term is only a mathematically defined parameter if the fundamental assumptions (indicated by ●) are violated. In this case, the estimated transport term also indicates the additional causes of the change in the observed particle number concentration that are not accounted for. Thus, the estimated transport term should be compared with other data and analyzing results (e.g., wind direction or atmospheric stability) while it is not reliable to determine the contribution of transport using the estimated transport term alone. Conversely, this method can possibly be

modified to estimate the contributions to the observed aerosol size distributions due to other reasons, e.g., to analyze the change in particle dilution rate in the plume of a primary emission source when the emission rate is relatively constant.

## 5 Conclusions

A population balance method is proposed to estimate the contribution of transport to the temporal evolutions of the observed aerosol size distributions during atmospheric new particle formation events. This method is based on the aerosol general

dynamic equation in the continuous form. The condensational growth rate is assumed to be independent of the particle diameter in the analyzed particle size range. According to the assumptions and approximations, we recommend that the analyzed particle sizes locate in the range from ~10 nm to ~50 nm, which is usually wide enough to analyze the influence of transport during a new particle formation event. The other reasons for the change in particle number concentration except for condensational growth, coagulation loss, and particle transport are assumed to be negligible. This method is used to analyze the new particle

formation events observed in South-East Tibet, Fukue Island, and urban Beijing. The contribution of transport to the observed aerosol size distributions at the South-East Tibet site was significant according to the analysis. The change in the cumulative

transport term was correlated with the changes in local wind speed and direction. We hypothesize that new particles were formed elsewhere and subsequently transported to the observation site by the valley breeze. Significant influence of transport is found during the new particle formation events at the Fukue Island site. The estimated contribution of transport was correlated with the movement of the air masses or the change in the local wind field. These tests indicated the feasibility of the

balance method to analyze the new particle formation events in relatively clean atmosphere. During an intense new particle formation event observed in urban Beijing, the estimated contribution of transport was negligible compared to coagulation scavenging and condensational growth, indicating that the observed event was a regional new particle formation event. However, the estimated coagulation loss term and the condensational growth term were approximately one magnitude larger than the rate of change of particle number concentration ($dN/dt$) in urban Beijing. Accordingly, the relative error of the

estimated transport term was sensitive to the uncertainties in the estimated coagulation loss rate and condensational grow rate.

**Data availability**

The Matlab scripts for the proposed balance method are available upon request.

**Author contributions**

RC and JJ conceived this study. RC, DY, LY, YF, XL, YL, LL, JH, YM, LW, JZ, TS, and JJ conducted the field campaign in
South-East Tibet. IC and TS conducted the field campaign at Fukue Island. RC, DY, YF, XL, JH, YM, JZ, and JJ conducted the field campaign in urban Beijing. RC analyzed data using the proposed balance method with contributions from JJ and all other co-authors. RC and JJ wrote the paper with contributions from all other co-authors.

**Competing interests**

The authors declare that they have no conflict of interest.

**Acknowledgement**

Financial support from the National Key R&D Program of China (2017YFC0209503) and the National Science Foundation of China (21521064, 41730106 & 91643201) is acknowledged. The authors gratefully acknowledge the NOAA Air Resources Laboratory (ARL) for the provision of the HYSPLIT transport and dispersion model and/or READY website (http://www.ready.noaa.gov) used in this study. We thank Prof. Peter McMurry at University of Minnesota for helpful
discussions on the assumptions and uncertainties of the proposed method.

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



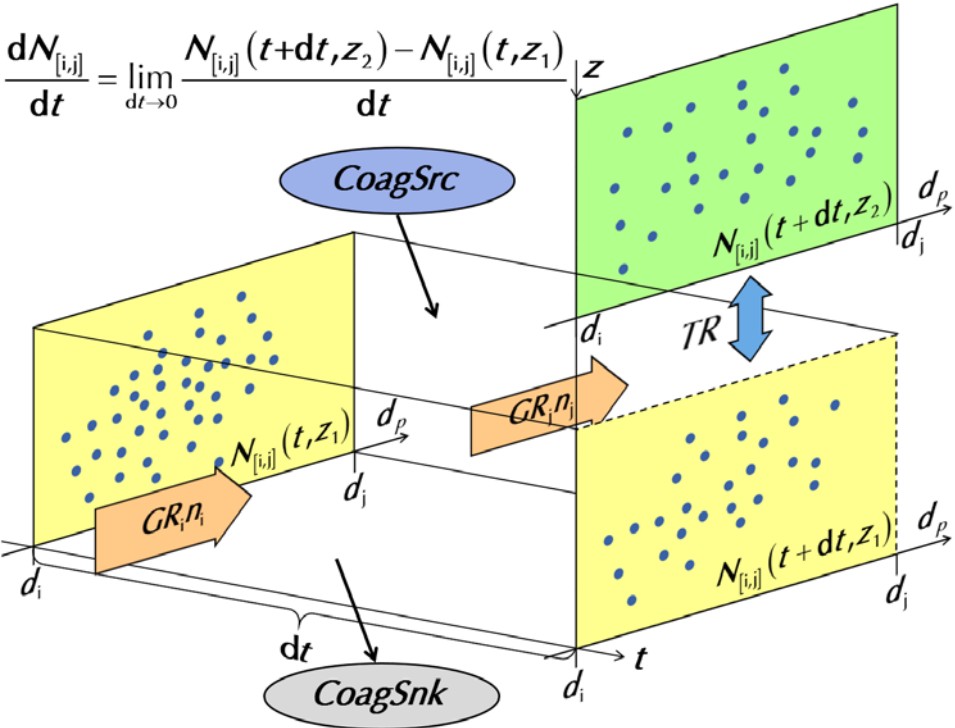

**Figure 1: The schematic to explain the definition of the transport term. The subscript [i,j] of *CoagSrc*, *CoagSnk*, and *TR* are omitted in the figure. The boxes with solid edge represent the observed particles at different time. The box with semi-dashed edge cannot be detected because observed aerosol population changes with time, however, their number concentration can been theoretically estimated using the general dynamic equation. The z axis denotes different aerosol populations. z corresponds to the Lagrangian positions if there is no mixing among particles at different spatial positions.**




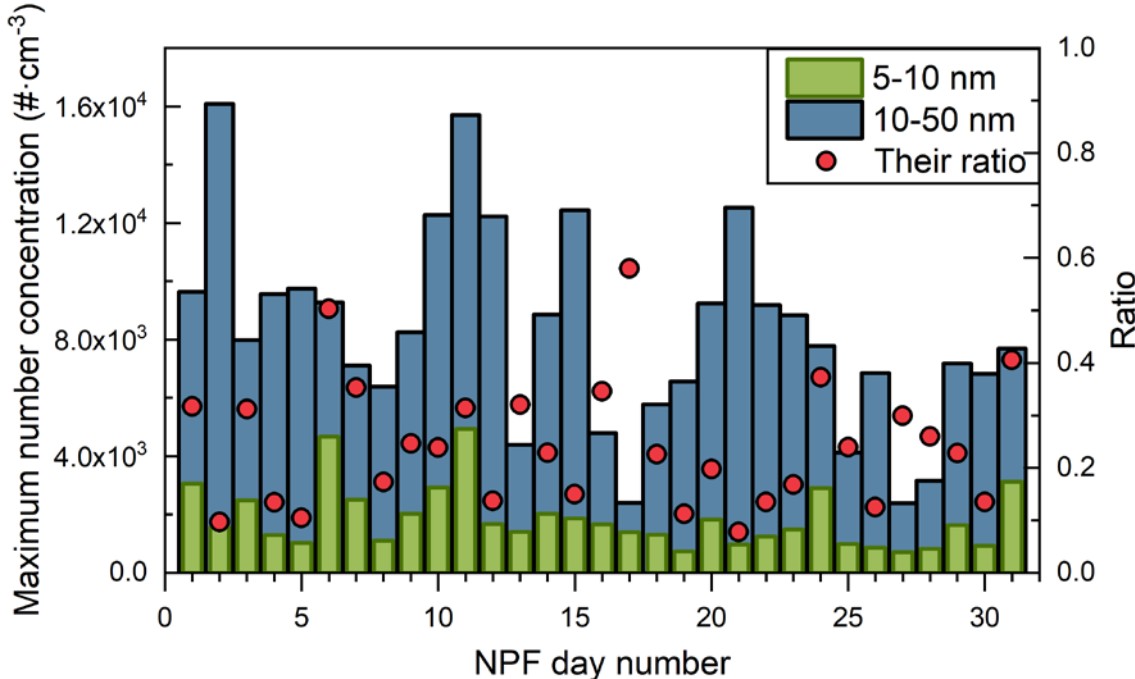

**Figure 2: The daily maximum number concentrations of 5-10 nm particles and 10-50 nm particles during the new particle formation period at the Lulang river valley. Note that the exact time corresponding to the two maximum values on each NPF day was not necessarily the same.**



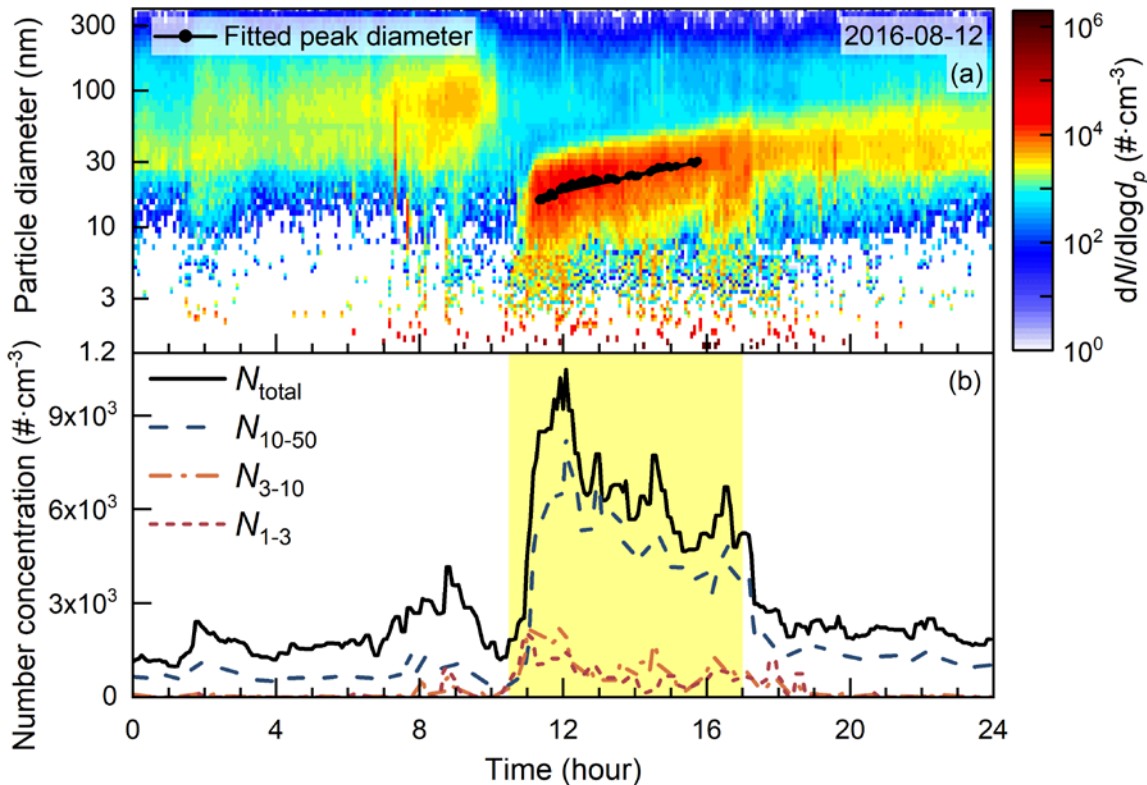

**Figure 3: The temporal evolution of the aerosol size distribution observed at the Lulang river valley on Aug. 12th, 2016. (a) The contour plot of aerosol size distributions. The black dots with the solid line are the peak diameters of the fitted lognormal distributions. The growth rate of the new particles was determined as the average slope of the fitted peak diameter over time. (b) The time series of the number concentration of 2 nm – 10 μm ($N_{total}$), 10-50 nm, 3-10 nm, and 1-3 nm (reported by the PSM) particles. The period of time governed by the valley breeze is shadowed with light yellow (see Fig. 4).**



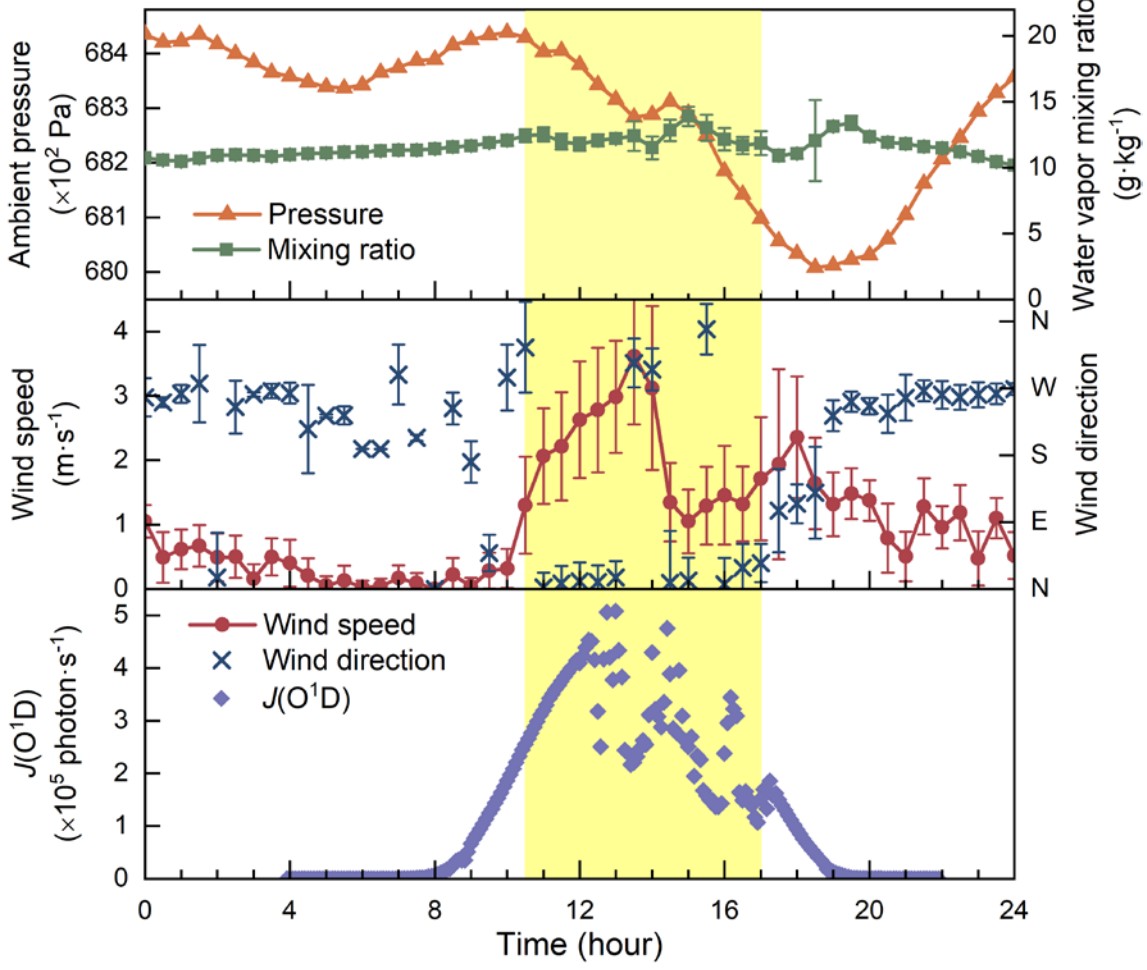

**Figure 4: The time series of atmospheric pressure, water vapor mixing ratio, wind speed, wind direction, and the photolysis frequencies of ozone, $J(O^1D)$, on Aug. 12$^{th}$, 2016 at the Lulang river valley. The period of time when wind came from north (indicating the valley breeze) is shadowed with light yellow. The error bars are the standard deviations.**



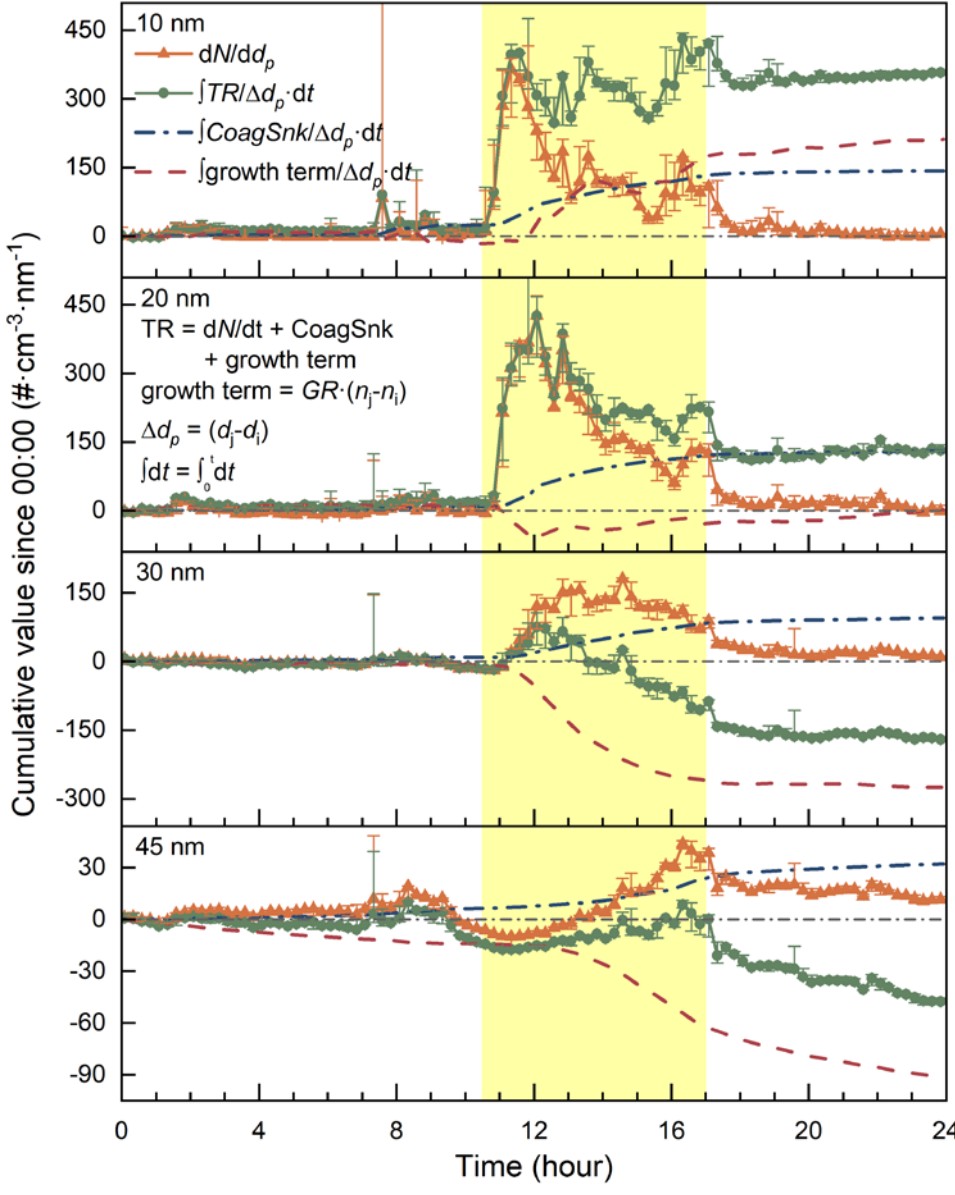

**Figure 5: The time series of** d$N$/d$d_p$ **(i.e., the cumulative** d$N$/d$t$**), the cumulative transport term, the cumulative** *CoagSnk* **and the cumulative growth term in Eq. 11 at four different diameters on Aug. 12th, 2016, at the Lulang river valley. Note that** d$N$/d$d_p$ **represented by yellow symbols and lines was zero at 0:00 according to Eq. 11. The** d$N$/d$d_p$ **and** *TR* **were averaged every 15 minutes, and the error bars indicate the maximum and minimum values in the corresponding time bins. The period of time when wind blew from north (indicating the valley breeze) is shadowed with light yellow.**



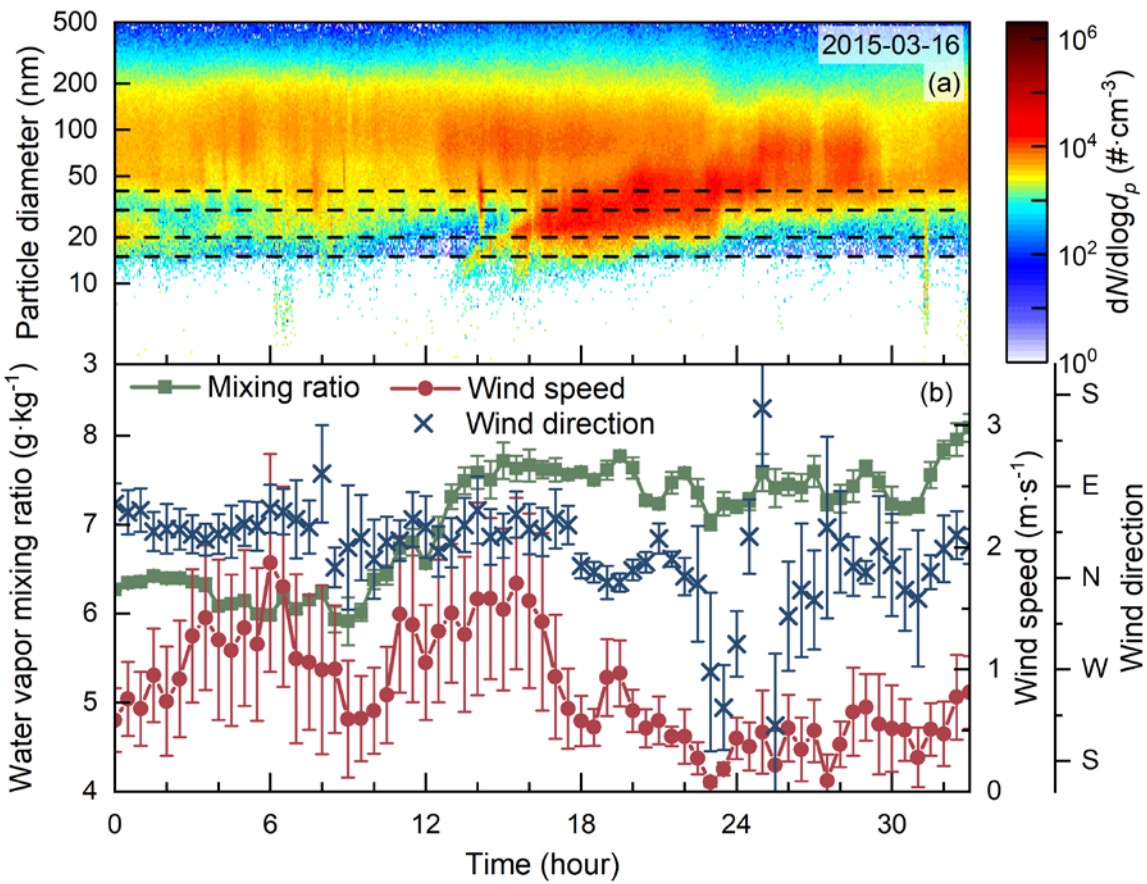

**Figure 6:** (a) The contour plot of aerosol size distributions of a typical type-B new particle formation event observed on Mar. 16th, 2015, at the Fukue Island site. The dashed lines indicates the analyzed particle diameters. (b) The time series of wind speed, wind direction, and water vapor mixing ratio. The error bars are the standard deviations.



**Figure 7: The time series of d$N$/dd$_p$ and the cumulative transport term on the Mar. 16-17, 2015, at the Fukue Island. The d$N$/dd$_p$ and cumulative transport term were averaged every 9 minutes. The error bars indicate the maximum and minimum values in the corresponding time bins.**





**Figure 8: (a) The contour plot of aerosol size distributions of a typical type-A new particle formation event observed on Mar. 2ⁿᵈ, 2016, at the Fukue Island site. The dashed lines indicates the analyzed particle diameters. (b) The time series of wind speed, wind direction, and water vapor mixing ratio. (c) The time series of the estimated cumulative transport terms for various sized particles.**




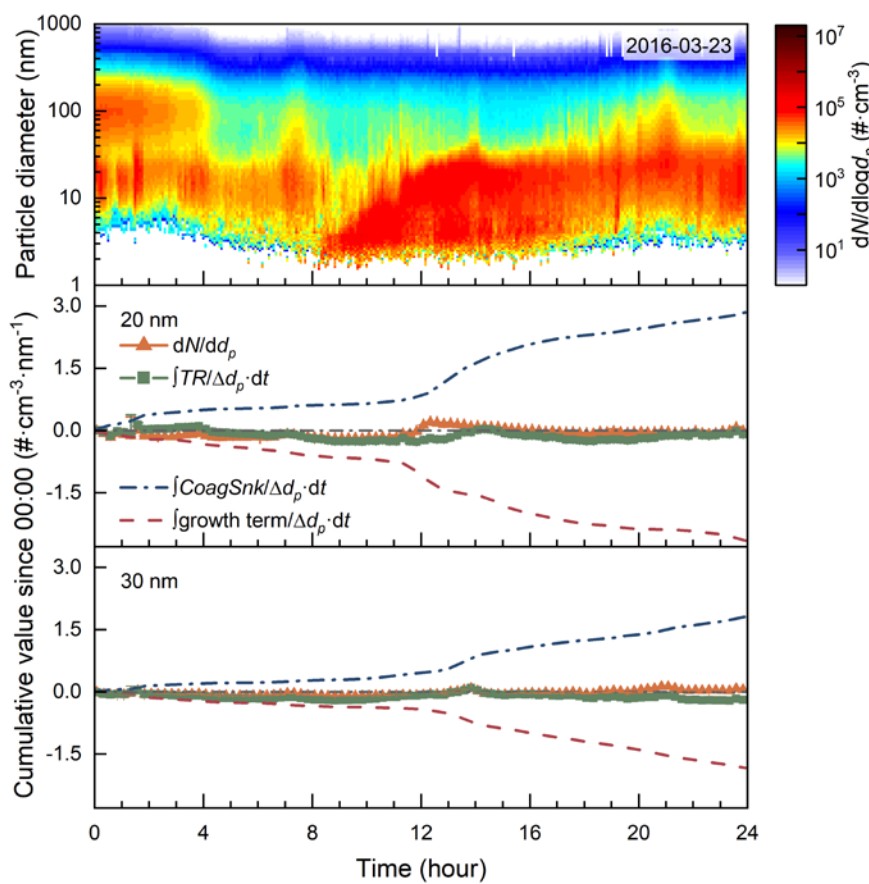

**Figure 9: The contour plot of aerosol size distribution and the time series of d$N$/d$d_p$, the cumulative transport term, the cumulative *CoagSnk* and the cumulative growth term on Mar. 23th, 2016, in urban Beijing. d$N$/d$d_p$ and the cumulative transport term were averaged every 15 minutes. The cumulative *CoagSrc* was estimated but not shown due to its negligible values.**

