# Peer review of "Estimating the influence of transport to aerosol size distributions during new particle formation events"

_Atmospheric Chemistry and Physics, 2018_

## Referee Comment (RC1) · Anonymous Referee #1 · 14 Sep 2018

This is a very well-written manuscript dealing with an important topic: the effect of transport on interpreting new particle formation (NPF) events using particle number size distribution (PNSD) measurements. Although it is well known that PNSDs are affected by inhomogeneities in measured air masses, no proper tools to take this into account in analyzing NPF event have been proposed so far. This manuscript addresses this topic. I have couple of issues that the authors could discuss a bit more in the paper and a few other relatively minor comments. After the revisions, the paper should definitely accepted for publication.

Main issues

[Figure]

The authors do not comment anything about the width of the considered size range [di, dj]. It is clear that there are both benefits and drawbacks of using either a narrower or broader size range. For example, a broad size range would worsen some of the assumptions stated on page 6, such as the influence of primary emissions and constancy of the particle growth rate (GR). A very narrow size range would probably cause more noise into some of the terms that influence the calculation of TR from equation 10. Please discuss shortly this issue in the paper.

As demonstrated by the equations derived in this paper, knowing the particle growth rate (GR) is needed to estimate the transport effect on NPF (the same concerns also calculating other relevant quantities related to NPF like the particle formation rate). The authors need to assume a constant GR to apply equation 10. I have a few comments related to this. First, the authors state on page 6 that constant GR is a good assumption over the size range 10 to 50 nm. This is only true if particles are growing by condensation of essentially non-volatile vapors. A number of studies have reported a strongly size-dependent GR in the sub-20 nm size range, and usually explained this feature either by more and more volatile organic vapors being able to condense onto particles as they get bigger. Furthermore, any contribution to GR from heterogeneous processes in growing particle would probably make GR not constant with particle size. This may be important, as e.g. Paasonen et al (2018, Atmos. Chem. Phys. 18, p. 12085) showed that, in long-term data from one measurement site, the average GR increased by about a factor 3 from the particle diameter of 10 nm to the diameter of 100 nm. Second, in cases where the transport effects are most important, it may either be very difficult to determine GR from measured PNSD data or, in case GR can be determined, it might not reflect the real GR of the measured particle population. The authors should bring up these issues and also comment shortly whether, and in which cases, they would cause problems in determining TR from equation 10.

Minor issues

In principle, all figures should be cited in a numerical order in the text. This is not the

case for Fig. 3a on line 14, page 6. However, in this case referring to figure 3a there is understandable as it is given as an example and then treated in more detail later in the text. To make this clear, it would replace "(see Fig. 3a)" on this line with "(see Fig. 3a in section 4.1)".

There is a large body of literature on liquid-phase reactions in aerosol particles after the study by McMurry and Wilson (1983) cited on line 5, page 7. I would recommend adding one or two more recent papers into here.

Grammatical issues:

page 2 line 24: . . ., no dramatic increase . . . was observed, . . .

page 2, line 4: . . . contribution . . .. to. . .

page 8, line 24: . . . at around 10:30 . . ..

page 10, line 30: . . . were formed. . .

---

## Referee Comment (RC2) · Anonymous Referee #2 · 26 Sep 2018

General comments:

This paper deals with an important topic for NPF measurement interpretation: estimating the effect of transport on the observed NPF events. The authors clearly explained how the population balance method can be used to evaluate whether the transport term is significant. Although some comments may be added to strengthen some points made in the paper, the paper is well written and the authors did not overstate their conclusions. I recommend this paper be accepted for publication after the authors address the two comments listed below as well as the issues brought up by the first referee.

Specific comments:

1. My major concern with the current method is the neglect of transport on apparent particle growth: 'the contribution of transport on apparent diameter is neglected in this study' (page 6, line 12). Although it is possible the effect of transport on apparent particle growth is indeed negligible, the opposite may also be true: the observed growth is due to particle growth elsewhere, and these grown particles are transported to the observation site. As pointed out by the authors, for the south-east Tibet observation, the correlation between local wind field and estimated transport terms suggest the particles may have been formed elsewhere (page 10, line 19). It is my understanding that the application of equation 10 requires the growth term to be decoupled from the transport term; while inferring growth from the peak diameter shift does not guarantee this decoupling. I suggest the authors add more comments on this.

2. For the urban Beijing observations, the particle number concentration are quite high. In this case, it is possible that cogulation contributes to the apparent particle growth in this. For this data set, it seems that the authors didn't apply correction techniques, e.g. the method given by Stoltzenburg et al.(2005), to account for the coagulation effect on particle growth. After accounting for coagulation, to what extent will the growth term shift?

Technical corrections:

page 1 line 28: ... in a relatively clean atmospheric environment...

page 9 line 23: ...at different particle diameters are correlated with...

page 11 line 30: ... in a relatively polluted environment...

———————————————

---

## Author Response (AR1)

**Responses to Reviewers' Comments on Manuscript ACPD-2018-844**

**(Estimating the influence of transport to aerosol size distributions during new particle formation events)**

We thank the reviewers for their comments to improve this manuscript. We have addressed the comments in the following paragraphs and made corresponding changes in the revised manuscript. Comments are shown as *blue italic text* followed by our responses. Changes are highlighted in the revised manuscript and shown as underlined text in the responses.

Reviewer #1:

*This is a very well-written manuscript dealing with an important topic: the effect of transport on interpreting new particle formation (NPF) events using particle number size distribution (PNSD) measurements. Although it is well known that PNSDs are affected by inhomogeneities in measured air masses, no proper tools to take this into account in analyzing NPF event have been proposed so far. This manuscript addresses this topic. I have couple of issues that the authors could discuss a bit more in the paper and a few other relatively minor comments. After the revisions, the paper should definitely accepted for publication.*

*Main issues*
*The authors do not comment anything about the width of the considered size range [di, dj]. It is clear that there are both benefits and drawbacks of using either a narrower or broader size range. For example, a broad size range would worsen some of the assumptions stated on page 6, such as the influence of primary emissions and constancy of the particle growth rate (GR). A very narrow size range would probably cause more noise into some of the terms that influence the calculation of TR from equation 10. Please discuss shortly this issue in the paper.*

Response: Thanks for the suggestion. We addressed the selection of the analyzed size range and the corresponding reason in the revised manuscript and supplementary materials. We also move Eq. 11 and the relevant paragraph on converting the transport term into the cumulative transport term to Section 2, because integrate the transport term with respect to time can also reduce the influence of random measurement uncertainty in addition to using a broad size range. The relevant paragraph is revised as:

"The estimated transport term usually needs to be properly smoothed to reduce the impact of the uncertainties in the estimated d$N_{[i,j]}$/d$t$, $CoagSnk_{[i,j]}$, and the growth term. Using a wide size range of the concerned size bin ([$d_i$, $d_j$]) can help to reduce measurement uncertainties. Alternatively, the transport term can be integrated with respect to time. The temporal evolution of particle number concentration (d$N_{[i,j]}$/d$t$) and the other terms in Eq. 10 can be converted into the temporal evolution...In addition to reducing the impact of the fluctuations in the observed aerosol size distributions due to measurement uncertainties, these transformations facilitate the comparison

among different size bins. In this study, we use a single measured size bin to analyze the contribution of transport to minimize the systematic error caused by the assumption of a size-independent *GR*. The representativeness of an analyzed size bin are tested by comparing to its adjacent size bins, as the example shown in Fig. S1."

*As demonstrated by the equations derived in this paper, knowing the particle growth rate (GR) is needed to estimate the transport effect on NPF (the same concerns also calculating other relevant quantities related to NPF like the particle formation rate). The authors need to assume a constant GR to apply equation 10. I have a few comments related to this. First, the authors state on page 6 that constant GR is a good assumption over the size range 10 to 50 nm. This is only true if particles are growing by condensation of essentially non-volatile vapors. A number of studies have reported a strongly size-dependent GR in the sub-20 nm size range, and usually explained this feature either by more and more volatile organic vapors being able to condense onto particles as they get bigger. Furthermore, any contribution to GR from heterogeneous processes in growing particle would probably make GR not constant with particle size. This may be important, as e.g. Paasonen et al (2018, Atmos. Chem. Phys. 18, p. 12085) showed that, in long-term data from one measurement site, the average GR increased by about a factor 3 from the particle diameter of 10 nm to the diameter of 100 nm. Second, in cases where the transport effects are most important, it may either be very difficult to determine GR from measured PNSD data or, in case GR can be determined, it might not reflect the real GR of the measured particle population. The authors should bring up these issues and also comment shortly whether, and in which cases, they would cause problems in determining TR from equation 10.*

Response: In the Theory section, we added:

"The accuracy of the estimated transport term is affected by the uncertainty of the estimated growth rate. Equation 10… Even in the recommended size range, 10-50 nm, particle growth rate may sometimes be size dependent due to the uptake of semi-volatile vapors (Paasonen et al., 2018). Accordingly, we recommend to use a narrow size range for estimating the transport term to minimize the potential systematic error caused by size-dependent growth rate. The change in particle diameter due to transport may sometimes contribute significantly to the estimated growth rate. For instance, a diameter shift in particle diameter due to a a sudden shift in wind direction may be mistaken as condensational growth if using a time-resolved growth rate. It is usually difficult to decouple the particle diameter shift due to transport and condensational growth rate because the contribution of transport is usually assumed negligible when estimating the growth rate. Accordingly, we recommend to determine growth rate via fitting particle peak diameter over a time range only when the wind speed and direction are relatively stable. Although this fitting method does not decouple the influence of transport, it may help to reduce uncertainties in the estimated growth rate. In addition, if there is a significant error in the method to estimate the growth rate (Li and McMurry, 2018), the error will also propagate into the estimated transport term."

In section 4.4 (Remarks on the feasibility of the balance method), we added "The errors in the estimated CoagSnk and the growth term also contribute to the uncertainties in the estimated transport term".

*Minor issues*
*In principle, all figures should be cited in a numerical order in the text. This is not the C2 case for Fig. 3a on line 14, page 6. However, in this case referring to figure 3a there is understandable as it is given as an example and then treated in more detail later in the text. To make this clear, it would replace "(see Fig. 3a)" on this line with "(see Fig. 3a in section 4.1)".*

Response: We revised "(see Fig. 3a)" as "(see Fig. 3a in section 4.1)".

*There is a large body of literature on liquid-phase reactions in aerosol particles after the study by McMurry and Wilson (1983) cited on line 5, page 7. I would recommend adding one or two more recent papers into here.*

Response: We added Moch et al. (2018) and Song et al. (2018). This sentence was revised as

"…liquid phase reactions are possibly an important mechanism for particle growth (e.g., McMurry and Wilson, 1983; Moch et al., 2018; Song et al., 2018), indicating…".

*Grammatical issues:*
*page 2 line 24: . . ., no dramatic increase . . . was observed, . . .*
*page 2, line 4: . . . contribution . . .. to. . .*
*page 8, line 24: . . . at around 10:30 . . ..*
*page 10, line 30: . . . were formed. . .*

Response: Thanks, corrected.

**Reviewer #2:**

*General comments:*
*This paper deals with an important topic for NPF measurement interpretation: estimating the effect of transport on the observed NPF events. The authors clearly explained how the population balance method can be used to evaluate whether the transport term is significant. Although some comments may be added to strengthen some points made in the paper, the paper is well written and the authors did not overstate their conclusions. I recommend this paper be accepted for publication after the authors address the two comments listed below as well as the issues brought up by the first referee.*

*Specific comments:*
*1. My major concern with the current method is the neglect of transport on apparent particle growth: 'the contribution of transport on apparent diameter is neglected in this study' (page 6, line 12). Although it is possible the effect of transport on apparent particle growth is indeed negligible, the opposite may also be true: the observed growth is due to particle growth elsewhere, and these grown particles are transported to the observation site. As pointed out by the authors, for the southeast Tibet observation, the correlation between local wind field and estimated transport terms suggest the particles may have been formed elsewhere (page 10, line 19). It is my understanding that the*

*application of equation 10 requires the growth term to be decoupled from the transport term; while inferring growth from the peak diameter shift does not guarantee this decoupling. I suggest the authors add more comments on this.*

Response: We discussed the uncertainty in the Theory section since the current method (Eq. 10) cannot decouple the contribution of transport to the observed peak diameter shift:

"The change in particle diameter due to transport may sometimes contribute significantly to the estimated growth rate. For instance, a diameter shift in particle diameter due to a a sudden shift in wind direction may be mistaken as condensational growth if using a time-resolved growth rate in Eq. 10. It is usually difficult to decouple the particle diameter shift due to transport and condensational growth rate because the contribution of transport is usually assumed negligible when estimating the growth rate. Accordingly, we recommend to determine growth rate via fitting particle peak diameter over a time range when the wind speed and direction are relatively stable. Although this fitting method does not decouple the influence of transport, it may help to reduce uncertainties in the estimated growth rate."

For the south-east Tibet observation, we added "The relatively constant cumulative transport terms of 10 nm and 20 nm particles between 12:00 and 14:00 indicate a relatively small contribution of transport to the observed shift in particle diameter and hence the estimated growth rate." The relatively constant cumulative transport term illustrated in Eq. 11 indicates an averagely near zero transport term in Eq. 10. Thus, transport may not be the reason of the observed particle diameter shift. For other specific events, the influence of transport on apparent particle growth may be significant and we think the newly added paragraph in the Theory section explains this problem.

*For the urban Beijing observations, the particle number concentration are quite high. In this case, it is possible that cogulation contributes to the apparent particle growth in this. For this data set, it seems that the authors didn't apply correction techniques, e.g. the method given by Stoltzenburg et al.(2005), to account for the coagulation effect on particle growth. After accounting for coagulation, to what extent will the growth term shift?*

Response: We corrected the coagulation contribution to the apparent growth rate, updated Fig. 9, and added "The contribution of coagulation to the estimated grow rate was corrected (Stoltzenburg et al., 2005)" in the main text and the caption of Fig, 9. The growth term decreases ~15% after correction. The cumulative transport term is still negligible compared to the cumulative *CoagSnk* and cumulative growth term after correction, as shown in Fig. R1.

[Figure]

Fig. R1: The time series of d$N$/d$d_p$, the cumulative transport term, the cumulative *CoagSnk* and the cumulative growth term of 30 nm particles before and after the correction of the coagulation contribution to particle growth.

We understand that the accuracy of the estimated growth rate is important to the estimated transport terms. Although coagulation contributes a minor proportion to particle growth during this event, it may be important during other events. Hence, we stated that "in relatively polluted atmospheric environment such as urban Beijing, however, the estimated the contribution of transport is sensitive to the uncertainties in CoagSnk and the growth term" in the original manuscript and listed "particle formation due to coagulation (*CoagSrc*[i,j]) is negligible compared to particle loss due to coagulation scavenging (*CoagSnk*[i,j]) and condensational growth (characterized using the growth term)" as an approximation of the current method.

*Technical corrections:*
*page 1 line 28: ... in a relatively clean atmospheric environment...*
*page 9 line 23: ...at different particle diameters are correlated with...*
*page 11 line 30: ... in a relatively polluted environment...*

Response: Thanks, corrected.

References

[revised manuscript text omitted]
_{[\mathrm{i},\mathrm{j}]}}{\mathrm{d}t} = GR_\mathrm{i}n_\mathrm{i} - GR_\mathrm{j}n_\mathrm{j} + CoagSrc_{[\mathrm{i},\mathrm{j}]} - CoagSnk_{[\mathrm{i},\mathrm{j}]} + TR_{[\mathrm{i},\mathrm{j}]},\tag{1}$$

where the subscripts i and j correspond to the specific particle diameters ($d_p$, in m) $d_\mathrm{i}$ and $d_\mathrm{j}$, respectively; $N_{[\mathrm{i},\mathrm{j}]}$ (in #·m$^{-3}$) is the number concentration of particles ranging from $d_\mathrm{i}$ to $d_\mathrm{j}$; $t$ is time (in s); $\mathrm{d}N/\mathrm{d}t$ characterizes the change in the observed particle number concentration (in #·m$^{-3}$·s$^{-1}$); $GR$ is the condensational growth rate (in m·s$^{-1}$) defined as $\mathrm{d}d_p/\mathrm{d}t$; $n$ is the aerosol size distribution function (in #·m$^{-4}$), $\mathrm{d}N/\mathrm{d}d_p$; $CoagSrc_{[\mathrm{i},\mathrm{j}]}$ and $CoagSnk_{[\mathrm{i,\,j}]}$ are the formation and loss rates due to coagulation for particles ranging from $d_\mathrm{i}$ to $d_\mathrm{j}$ (in #·m$^{-3}$·s$^{-1}$), respectively; and $TR_{[\mathrm{i},\mathrm{j}]}$ is the newly introduced transport term (in #·m$^{-3}$·s$^{-1}$, to be explained below). The terms on the right-hand side of Eq. 1 correspond to the five processes leading to the change in observed particle number concentration: particle condensational growth into and out of the size range, formation and scavenging due to coagulation, and the contribution of transport. The theoretical expressions for $CoagSrc_{[\mathrm{i},\mathrm{j}]}$ and $CoagSnk_{[\mathrm{i},\mathrm{j}]}$ in the integral form are shown in Eq. 2 and Eq. 3, respectively (Kuang et al., 2012).

$$CoagSrc_{[\mathrm{i},\mathrm{j}]} = \frac{1}{2}\int_{d_\mathrm{i}}^{d_\mathrm{j}}\int_{0}^{d_y}\beta_{(x,\bar{x})}n_x n_{\bar{x}}\frac{d_y^{\,2}}{d_{\bar{x}}^{\,2}}\mathrm{d}d_x\mathrm{d}d_y\tag{2}$$

$$CoagSnk_{[\mathrm{i},\mathrm{j}]} = \int_{d_\mathrm{i}}^{d_\mathrm{j}}\int_{0}^{+\infty}\beta_{(x,y)}n_x n_y\mathrm{d}d_x\mathrm{d}d_y\tag{3}$$

$\beta_{(x,\,y)}$ is the coagulation coefficient (m$^3$·s$^{-1}$) for particles with the diameter $d_\mathrm{x}$ and those with the diameter $d_\mathrm{y}$. The value of $\beta_{(x,\,y)}$ can be estimated using the Fuchs' formula (Eq. 13.56, Seinfeld and Pandis, 2006). $d_\mathrm{x}$ and $d_\mathrm{y}$ are variables representing particle

diameters determined by the limits of integration. $\bar{x}$ is the subscript of $d_{\bar{x}}$ and $d_{\bar{x}}$ is defined by $d_x^3 + d_{\bar{x}}^3 = d_y^3$. $n_x$, $n_y$, and $n_{\bar{x}}$ are the aerosol size distribution functions (d$N$/d$d_p$) at $d_x$, $d_y$, and $d_{\bar{x}}$, respectively.

The transport term in Eq. 1, $TR_{[i,j]}$, characterizes the contribution of transport to the observed d$N_{[i,j]}$/d$t$. However, the physical meanings of $TR_{[i,j]}$ and d$N_{[i,j]}$/d$t$ should be specially clarified. When aerosol is well mixed, i.e., there is no difference among

5     the aerosol size distributions at different spatial positions, the observed d$N_{[i,j]}$/d$t$ ==is only affected by coagulation and condensational growth into and out of the size range==. However, there is no absolute homogeneity of aerosols in the real ambient air. Since the air is constantly flowing whereas the sampling inlet for the instrument is usually fixed at a specific position, the observed aerosol size distributions at different time actually correspond to different aerosol populations. As shown in Fig. 1, the observed $N_{[i,j]}$ at time $t$ is $N_{[i,j]}(t, z_1)$ and $z_1$ denotes the detected aerosol population at $t$. For the sake of illustration, we

10     may assume that the aerosols at different spatial positions do not mix with each other and hence $z_1$ can be related to the Lagrangian position of the observed aerosol population. After a short time interval, d$t$, we assume that the observed particle number concentration is $N_{[i,j]}(t+\mathrm{d}t, z_2)$ where $z_2$ indicates that the detected aerosol population has changed due to transport. The measured time rate of change of particle number concentration, d$N_{[i,j]}$/d$t$ is the ==synergistic== result of changes in both time and fluid position, which can be mathematically described as:

$$\frac{\mathrm{d}N_{[i,j]}}{\mathrm{d}t} = \lim_{\mathrm{d}t \to 0} \frac{N_{[i,j]}(t+\mathrm{d}t, z_2) - N_{[i,j]}(t, z_1)}{\mathrm{d}t}. \tag{4}$$

At time $t+$d$t$, the number concentration of aerosols at ==the initial== position $z_1$ should become $N_{[i,j]}(t+\mathrm{d}t, z_1)$. The value of $N_{[i,j]}(t+\mathrm{d}t, z_1)$ can be theoretically estimated using the general dynamic equation:

$$\lim_{\mathrm{d}t \to 0} \frac{N_{[i,j]}(t+\mathrm{d}t, z_1) - N_{[i,j]}(t, z_1)}{\mathrm{d}t} = GR_i n_i - GR_j n_j + CoagSrc_{[i,j]} - CoagSnk_{[i,j]}, \tag{5}$$

Equations 1, 4, and 5 indicate that $TR_{[i,j]}$ essentially characterizes the difference between the measured and the theoretically

20     predicted aerosol concentrations, i.e., the difference between the two aerosol populations ==at== $z_1$ and $z_2$, as shown in Eq. 6.

$$TR_{[i,j]} = \lim_{\mathrm{d}t \to 0} \frac{N_{[i,j]}(t+\mathrm{d}t, z_2) - N_{[i,j]}(t+\mathrm{d}t, z_1)}{\mathrm{d}t} \
[revised manuscript text omitted]